# Physiological and Biochemical Parameters of Salinity Resistance of Three Durum Wheat Genotypes

**DOI:** 10.3390/ijms23158397

**Published:** 2022-07-29

**Authors:** Jakub Pastuszak, Michał Dziurka, Marta Hornyák, Anna Szczerba, Przemysław Kopeć, Agnieszka Płażek

**Affiliations:** 1Department of Plant Breeding, Physiology and Seed Science, University of Agriculture, Podłużna 3, 30-239 Kraków, Poland; anna.szczerba21@gmail.com; 2The Franciszek Górski Institute of Plant Physiology, Polish Academy of Sciences, Niezapominajek 21, 30-239 Kraków, Poland; m.dziurka@ifr-pan.edu.pl (M.D.); p.kopec@ifr-pan.edu.pl (P.K.); 3Władysław Szafer Institute of Botany, Polish Academy of Sciences, Lubicz 46, 31-512 Kraków, Poland; marta.hornyak@botany.pl

**Keywords:** antioxidant enzyme activity, chlorophyll fluorescence, durum wheat, gas exchange, germination vigor, salinity

## Abstract

The area of farming lands affected by increasing soil salinity is growing significantly worldwide. For this reason, breeding works are conducted to improve the salinity tolerance of important crop species. The goal of the present study was to indicate physiological or biochemical parameters characterizing three durum wheat accessions with various tolerance to salinity. The study was carried out on germinating seeds and mature plants of a Polish SMH87 line, an Australian cultivar ‘Tamaroi’ (salt-sensitive), and the BC_5_Nax_2_ line (salt-tolerant) exposed to 0–150 mM NaCl. Germination parameters, electrolyte leakage (EL), and salt susceptibility index were determined in the germinating caryopses, whereas photosynthetic parameters, carbohydrate and phenolic content, antioxidant activity as well as yield were measured in fully developed plants. The parameters that most differentiated the examined accessions in the germination phase were the percentage of germinating seeds (PGS) and germination vigor (*Vi*). In the fully developed plants, parameters included whether the plants had the maximum efficiency of the water-splitting reaction on the donor side of photosystem II (PSII)–F_v_/F_0_, energy dissipation from PSII–DI_o_/CS_m_, and the content of photosynthetic pigments and hydrogen peroxide, which differentiated studied genotypes in terms of salinity tolerance degree. Salinity has a negative impact on grain yield by reducing the number of seeds per spike and the mass of one thousand seeds (MTS), which can be used as the most suitable parameter for determining tolerance to salinity stress. The most salt-tolerant BC_5_Nax_2_ line was characterized by the highest PGS, and *Vi* for NaCl concentration of 100–150 mM, content of chlorophyll *a*, *b*, carotenoids, and also MTS at all applied salt concentrations as compared with the other accessions. The most salt-sensitive cv. ‘Tamaroi’ demonstrated higher H_2_O_2_ concentration which proves considerable oxidative damage caused by salinity stress. Mentioned parameters can be helpful for breeders in the selection of genotypes the most resistant to this stress.

## 1. Introduction

The problem of soil salinity is important due to the increased demand for food for the global population, growing every year. The main anthropogenic factors causing soil salinity are the intensification of agriculture, and inappropriate agronomic practices, which include the use of high doses of artificial fertilizers, inadequate irrigation, and deforestation [1,2,3]. Around 20% of agricultural lands are at risk of increased soil salinity. Rapid climate changes cause long-term droughts that affect more and more areas in Europe. This situation forces farmers to more frequently irrigate their crops with only partly desalinated water, which at growing air temperatures increases the salinity of the upper soil layers [4]. Sodium ions (Na^+^) in the soil decrease water osmotic potential, which in consequence reduces water uptake capacity by imbibed seeds and roots of plants cultivated in saline areas [5]. The Na^+^ and chlorine (Cl^−^) ions exert a toxic effect on plants and induce osmotic and ionic stress. The ionic stress limits the uptake of potassium (K^+^), calcium (Ca^2+^), magnesium (Mg^2+^), and nitrate (NO_3_^−^) ions important for plant functioning [6,7]. A high concentration of Na^+^ in the soil complex can lead to many permanent disorders at every stage of plant development. Salinity stress worsens numerous yield parameters [8,9,10]. The increased amount of toxic Na^+^ in the soil solution damages cell membranes in germinating seeds [11]. It also limits the biosynthesis of all chlorophyll fractions, lowers photosynthesis efficiency and stomatal closure, and has a negative impact on gas exchange [12,13]. An increased accumulation of reactive oxygen species (ROS) is also observed during salinity stress. High amounts of ROS evoke lipid peroxidation leading, i.e., to increased membrane fluidity and permeability, denaturation of DNA, changes in hormonal profile, and enzyme inactivation [14,15,16]. 

Osmotic stress can be evoked by not only salinity but also by drought and heavy metal ions. These stress factors change morphological traits causing the reduction in leaf size and vegetative growth, a decline in photosynthesis rate, stomatal conductance, and alter stem anatomical features [17,18,19,20,21]. To counteract the negative effects of direct exposure to salinity, plants have developed internal defense mechanisms. There are two types of mechanisms. The first is named the ‘avoidance strategy’, and it consists of avoiding salt stress by creating barriers that prevent the penetration of toxic ions into the plant. The second mechanism involves ‘salinity tolerance’. It is based on the development of intracellular mechanisms that minimize damage and make it possible to repair the negative consequences of stress [22]. Cellular mechanisms of resistance to salinity regulate ion transport. This is possible due to the Na^+^/H^+^ antiports, proton pumps, and ion channels. The presence and participation of membrane transporters are significant in order to maintain intracellular homeostasis [23,24,25]. To maintain a low concentration of toxic Na^+^ inside the cytoplasm, active transport of the ions outside the cell through the plasmalemma and their compartmentalization in the vacuole is necessary [4,24,26]. 

The defense mechanism against ROS is based on the production of enzymatic (catalase, peroxidases, superoxide dismutase) and non-enzymatic (ascorbic acid, glutathione, carotenoids, anthocyanins) antioxidants [27,28]. The first discovered and described antioxidant enzyme was catalase (CAT). Its essential role is to remove the excess H_2_O_2_ and degrade it in the cellular organelles during, e.g., fatty acid oxidation or photorespiratory oxidation [29,30]. Peroxidase (POX) also contributes to H_2_O_2_ removal from different cellular organelles, especially from the cell wall. POX utilizes apoplastic H_2_O_2_ in the lignification process [31,32]. Superoxide dismutase (SOD) is considered to be the first line of defense against oxidative stress in plants. The pivotal role of SOD is to catalyze the dismutation of O_2_^•−^ (superoxide) and HO_2_^•^ (hydroperoxide radical) to H_2_O_2_ and H_2_O and to maintain redox balance in the defense response of plants exposed to stress [33]. Hydrogen peroxide plays a signaling role that may facilitate plant response to various stimuli in plant cells and is involved in cellular signaling transduction pathways and gene expression modulations [34].

The cereal most commonly used in the food industry is durum wheat (*Triticum turgidum* L. subsp. *durum* (Desf.) Husn.). It is considered to be the tenth most important and cultivated cereal around the world. It is estimated that the global durum wheat production reaches up to 40 million tons. The world’s largest producers are Canada, America, and Turkey. In Europe, it is cultivated in the countries of the Mediterranean region [35]. Recent years have brought increased interest in durum wheat cultivation in Poland. The widespread use of durum wheat grains includes pasta production, and to a lesser extent bread and groats. The grain is characterized by a high content of grain protein and fiber, a low glycemic index, and a high level of vitamins and other valuable micronutrients [36,37]. Durum wheat is more sensitive to salinity stress than common wheat. In comparison with durum wheat, common wheat has well-developed mechanisms of salinity tolerance consisting of the excretion of salt ions from the cytosol [4]. Genetic analysis of many common wheat genotypes has identified one ‘Line 149’ which has two major Na^+^ exclusion loci named *Nax_1_* and *Nax_2_*. According to Huang et al. [38], some members of the HKT family (high-affinity K^+^ transporter) as sodium transporters play an important role in the regulation of Na^+^ content in the roots and shoots. HKT transporters appear important in the control of Na^+^ transport in bread wheat and may also transport sodium contributing to salt tolerance in durum wheat. The function of the *Nax_1_* gene is to remove Na^+^ from the xylem in the roots and lower parts of the leaves and leaf sheaths, whereas *Nax_2_* removes sodium ions from the xylem only in the roots [39,40]. The locations of the *Nax_1_* (chromosome 2A) and *Nax_2_* (chromosome 5A) genes were confirmed by quantitative trait locus (QTL) analysis and identified by fine-mapping the Na^+^ transporter from the *HKT* gene families—*HKT7* for *Nax_1_* and *HKT8* for *Nax_2_* [38,41]. Ibrahimowa et al. [21] studied the response of two *T. aestivum* genotypes differing in terms of salinity tolerance. These authors observed an increased expression level of the *TaHKT1;5* genes in the roots of the salt-sensitive genotype, and its decrease in the salt-tolerant one. 

The research hypothesis of our work was that it will be possible to choose physiological or biochemical markers of salinity tolerance useful for selection among the parameters studied in our paper. The research material consisted of two well-characterized Australian spring durum wheat accessions: the cv. ‘Tamaroi’ was found to be very sensitive to a high concentration of NaCl, whereas the BC_5_Nax_2_ line was created by crossing the cv. ‘Tamaroi’ with the 149 line, carrying the *TmHKT1;5* gene responsible for preventing the transport of sodium ions from the roots to shoots [42]. Additionally, the Polish SMH87 line was used, which was not examined in terms of salinity tolerance, so far. Until now, in Poland, the research on the resistance of durum wheat to salinity was conducted by our team, but on other genotypes [43]. The aim of the present work was to indicate the most important physiological or biochemical parameters shaping the durum wheat tolerance to salinity. The study was carried out on three accessions: Polish line SMH87 and two Australian genotypes: cv. ‘Tamaroi’ and the BC_5_Nax_2_ line. We performed two experiments: a laboratory test on germinating seeds exposed to 0, 50, 100, 125, and 150 mM NaCl (Experiment 1), and analyses performed on plants grown in soil watered with a saline solution containing 0, 100, 125, and 150 mM NaCl (Experiment 2). In Experiment 1, the responses of the studied durum wheat accessions to salinity were evaluated by determination of the percentage of germinating seeds, germinating vigor, coleoptile length, salt susceptibility index, and electrolyte leakage from germinating seeds. Experiment 2 involved the analyses of the kinetics of chlorophyll *a* fluorescence, gas exchange parameters, content of chlorophyll *a* and *b*, carotenoids, total soluble carbohydrates, total phenolic compounds, and cell-wall bound phenolics, the activity of superoxide dismutase, catalase, peroxidase, hydrogen peroxide content, and finally the evaluation of yield parameters. 

## 2. Results

### 2.1. Experiment 1

#### 2.1.1. Percentage of Germinated Seeds (PGS), Germination Vigor Index (*Vi*), Coleoptile Length (CL) and Salt Susceptibility Index (SSI)

The seeds were germinated in the presence of 50, 100, 120, and 150 mM NaCl. The percentage of germinated seeds (PGS) was calculated two days after sowing. The highest percentage of germinating seeds at all NaCl doses was seen in the BC_5_Nax_2_ line (Figure 1A). NaCl treatment significantly reduced the germination in the SMH87 line and cv. ‘Tamaroi’ at all applied doses. The lowest dose of NaCl (50 mM) did not reduce PGS only in the BC_5_Nax_2_ line. In the SMH87 plants, the lowest PGS was obtained at 125 mM NaCl, and it was by 13.9% lower than that of the control. A significantly lower PGS was observed in cv. ‘Tamaroi’ treated with 125 and 150 mM NaCl, and it was lower by respectively 21 and 26% than in the control. In the case of the BC_5_Nax_2_ line, 125 and 150 mM significantly reduced PGS by 11 and 10%, respectively. Moreover, treatments with 125 and 150 mM NaCl increased seed ability to germinate in the case of BC_5_Nax_2_, as compared with the other studied accessions. 

The vigor of germinating seeds (*Vi*) was compatible with the results obtained for the PGS (Figure 1B). Moreover, the lowest NaCl dose (50 mM) reduced *Vi* of the SMH87 line and cv. ‘Tamaroi’, whereas *Vi* did not change in plants of the BC_5_Nax_2_ line. The greatest reduction in *Vi* was observed in cv. ‘Tamaroi’, where NaCl at 125 and 150 mM reduced the parameter by 65 and 66%, respectively, as compared with the control. In SMH87 and BC_5_Nax_2_ seeds, 125 and 150 mM NaCl reduced *Vi* by over 58 and 48%, respectively, as compared with the control. 

With increasing salinity, the coleoptile length (CL) decreased in all studied accessions (Figure 1C). At 50 mM NaCl, a significant reduction by more than 12% versus the control coleoptiles was observed in cv. ‘Tamaroi’ and the BC_5_Nax_2_ line. The highest NaCl dose had a significant effect on the CL reduction in all studied accessions, whereas the greatest CL reduction (80%) was noted in cv. ‘Tamaroi’.

Salt susceptibility index (SSI) increased with growing NaCl doses in all accessions (Figure 1D). The treatment with 50 mM NaCl had a similar effect on SSI in the SMH87 and BC_5_Nax_2_ lines. Their SSI was elevated by over 12%, whereas in cv. ‘Tamaroi’ it rose by 20%. The highest SSI was observed in cv. ‘Tamaroi’ at 125 and 150 mM NaCl. At 125 mM NaCl, the SMH87 line showed lower sensitivity to salinity than the other genotypes, whereas the BC_5_Nax_2_ line turned out the least sensitive to salinity (59.9%) at 150 mM NaCl.

#### 2.1.2. Electrolyte Leakage (EL)

All NaCl treatments enhanced electrolyte leakage (EL) from germinating seeds (Figure 2). Even the lowest NaCl dose increased EL by more than 50% for each accession. The highest EL was observed in BC_5_Nax_2_ seeds at 150 mM NaCl. It was 6.2-fold higher than in the control. In the SMH87 line and cv. ‘Tamaroi’, the EL increased by 5.1 and 4.6 times, respectively.

#### 2.1.3. Correlation Analysis

The CL correlated negatively with the SSI and EL (Table 1). A positive correlation between CL and PGS and *Vi* was found. SSI positively correlated with EL, whereas a negative correlation was obtained between PGS and *Vi*. A negative correlation between EL and PGS, and between EL and *Vi* was observed. A positive correlation was also found between PGS and *Vi*.

### 2.2. Experiment 2

#### 2.2.1. Measurements of Chlorophyll *a* Fluorescence (ChlF)

The energy absorbed by the antennas (ABS/CS_m_) was different for the studied accessions exposed to salinity (Figure 3A). In the SMH87 line, the ABS/CS_m_ decreased with increasing NaCl concentration. In cv. ‘Tamaroi’ plants this parameter increased at 100 and 125 mM NaCl, respectively, and at 150 mM NaCl, it decreased to the control value. The BC_5_Nax_2_ line was characterized by the highest ABS/CS_m_ at 100 mM, and the lowest at 150 mM. The excitation energy trapped in PSII (TR_o_/CS_m_) was different for each studied genotype (Figure 3B). The highest applied salinity dose (150 mM) significantly decreased TRo/CS_m_ in the SMH87 and BC_5_Nax_2_ lines. In cv. ‘Tamaroi’ plants, such changes were not observed. The highest TR_o_/CS_m_ value was recorded for the BC_5_Nax_2_ line at 100 mM NaCl. The highest value of energy used for electron transport (ET_o_/CS_m_) was observed in all genotypes at 100 and 125 mM NaCl (Figure 3C). The lowest value of this parameter was detected at 150 mM NaCl in the SMH87 and BC_5_Nax_2_ lines. In cv. ‘Tamaroi’, this salinity dose did not change the ET_o_/CS_m_ values as compared with the control. The energy dissipation from PSII (DI_o_/CS_m_) did not change under any of the NaCl treatments in cv. ‘Tamaroi’ and BC_5_Nax_2_ plants (Figure 3D). In the SMH87 line, 100 and 125 mM NaCl decreased the value of this parameter, whereas 150 mM NaCl evoked its drastic increase. The number of active reaction centers (RC/CS_m_) increased under 100 mM NaCl in all studied genotypes (Figure 3E). The higher doses of NaCl lowered the RC/CS_m_ value in all studied accessions but in the ‘Tamaroi’ and BC_5_Nax_2_ plants at 150 mM NaCl the number of reaction centers was lower than that of the control. The same type of changes under salinity was observed for the performance index (PI) (Figure 3F). The highest value of PI was seen at 125 mM NaCl in SMH87 plants, whereas the lowest was in the BC_5_Nax_2_ line at 150 mM NaCl. The most pronounced effect of salinity on the maximum efficiency of water-splitting reaction of the donor side of PSII (F_v_/F_0_) was noted in SMH87 plants at 150 mM NaCl (Figure 3G). In the case of other accessions, this concentration also reduced the value of this parameter. 

#### 2.2.2. Measurements of Gas Exchange Parameters

Salinity had a significant impact on all gas exchange parameters (Figure 4). The net photosynthesis rate (*A*) was changed in all studied accessions under increasing salinity (Figure 4A). The increase in net photosynthesis rate in the SMH87 line was observed under all salt treatments, and it was over two-fold higher than in the control. In cv. ‘Tamaroi’ and the BC_5_Nax_2_ line, 100 mM NaCl diminished the net photosynthesis rate over two times. The increasing salinity boosted A significantly in cv. ‘Tamaroi’ plants. The highest photosynthetic efficiency of this cultivar was observed at 150 mM NaCl. In BC_5_Nax_2_ plants at 125 and 150 mM NaCl, this parameter was lower than in the control ones. 

Along with increased NaCl treatment, the intracellular CO_2_ concentration (*C_i_*) decreased in the SMH87 line (Figure 4B). Treatment with 125 and 150 mM NaCl significantly reduced the *C_i_* in cv. ‘Tamaroi’ and the BC_5_Nax_2_ line as compared with the control. In BC_5_Nax_2_ plants, *C_i_* was the highest at 100 mM NaCl.

In the SMH87 line, the highest transpiration rate (*E*) was observed in plants grown in the presence of 100 and 150 mM NaCl, and it was respectively 12% and 9% higher than in the control. NaCl at 125 mM reduced *E* by 23%. In cv. ‘Tamaroi’ and the BC_5_Nax_2_ line, the highest transpiration rate was observed in the control plants. In these genotypes, the parameter decreased with increasing soil salinity. In cv. ‘Tamaroi’, the concentration of 100 mM NaCl reduced *E* by 45%, whereas 125 and 150 mM NaCl reduced the parameter by 53% and 58%, respectively, as compared with the control. In the BC_5_Nax_2_ line, the transpiration rate decreased significantly by 35% and 41% at respectively 100 and 125 mM NaCl, and at 150 mM NaCl, it was 28% lower than in the control. In the SMH87 plants, the stomatal conductance (*g_s_*) increased at 100 mM NaCl but it was limited at 125 and 150 mM NaCl. In cv. ‘Tamaroi’ and the BC_5_Nax_2_ line, the changes in the stomatal conductance followed a similar pattern as for the transpiration rate, and *g_s_* was the highest in the control plants.

#### 2.2.3. Chlorophyll (*a*, *b*) and Carotenoid (Car) Content

The content of chlorophyll *a* (Chl *a*) differed under salt treatments in each accession (Figure 5A). In the SMH87 line, an increase in Chl *a* content was observed at 100 and 125 mM NaCl as compared with the control. In cv. ‘Tamaroi’, growing salinity reduced Chl *a* content. A decrease in Chl *a* was observed in the BC_5_Nax_2_ line at 100 and 125 mM NaCl versus the control, whereas at 150 mM NaCl a slight rise by 8% was detected. The SMH87 control plants and those grown at 125 mM NaCl showed comparable content of Chl *b* (Figure 5B). The doses of 100 and 150 mM NaCl reduced its content by 14% and 12%, respectively. In cv. ‘Tamaroi’, the increased soil salinity curbed Chl *b* content. In the BC_5_Nax_2_ line, treatments with 100 and 150 mM NaCl elevated Chl *b* content by 10% and 21%, respectively. A reduced carotenoid content (Car) was observed in SMH87 plants at 100 and 150 mM NaCl, as compared with the control (Figure 5C). In cv. ‘Tamaroi’, increasing salinity drastically reduced Car content, whereas an opposite effect was observed in BC_5_Nax_2_ plants, where 150 mM NaCl increased this pigment level by 11% as compared with the control.

#### 2.2.4. Total Soluble Carbohydrate (TSC) Content

In all accessions, increased salinity significantly enhanced TSC content (Figure 5D). In SMH87 plants, the highest TSC content was observed at 150 mM NaCl, and it was 46% greater than in the control plants. In cv. ‘Tamaroi’, increasing salinity resulted in a TSC spike, and its highest content was noted at 100 mM NaCl (it was 92% higher than in the control). In the BC_5_Nax_2_ line, the highest content of TSC was observed at 125 mM NaCl, and it was 70% greater than in the control. 

#### 2.2.5. Total Phenolic Compound (TPC) Content

The lowest amount of TPC was noted in the leaves of control SMH87 plants (Figure 5E). Increasing soil salinity resulted in a gradual decline in TPC in all studied genotypes. At 100 mM NaCl, TPC decreased in the SMH87 line by 22%, in cv. ‘Tamaroi’ by 17%, and in the BC_5_Nax_2_ line by 22% in relation to the control. In all accessions, the lowest TPC in the leaves was found at the highest salt concentration. 

#### 2.2.6. Cell Wall-Bound Phenolic (CWP) Content

The increasing salinity affected CWP content in the leaves of each accession under study (Figure 5F). Line SMH87 showed no changes in CWP content at 100 and 125 mM NaCl, whereas at 150 mM NaCl a slight increase, albeit significant, was found. In the leaves of other plants, CWP amount varied non-specifically under different salt concentrations. The highest content of CWP in cv. ‘Tamaroi’ was recorded at 100 mM NaCl, whereas in the BC_5_Nax_2_ line at 125 mM NaCl. 

#### 2.2.7. Superoxide Dismutase (SOD) Activity

The highest SOD activity was observed in SMH87 plants treated with 125 mM NaCl, and it was 22% higher than the control value (Figure 6C). In the case of cv. ‘Tamaroi’ and the BC_5_Nax_2_ line, the increased salinity declined the activity of this enzyme and it was the lowest at 150 mM NaCl.

#### 2.2.8. Catalase (CAT) Activity

In all studied genotypes, increasing salinity enhanced the activity of CAT as compared with the control (Figure 6A). In SMH87 plants, treatment with 125 mM NaCl resulted in the highest CAT activity which was 2.7 times higher than in the control. In cv. ‘Tamaroi’, the highest activity of this enzyme was observed at 125 and 150 mM NaCl, and it was over 3-fold higher than in the control plants. In the BC_5_Nax_2_ line, the highest CAT activity was detected in plants treated with 125 mM NaCl, and it was 2.8-fold higher than in the control plants.

#### 2.2.9. Peroxidase (POX) Activity

Along with increasing salinity, POX activity changed non-specifically in all plants under the study. The 125 mM NaCl treatment significantly enhanced POX activity in SMH87 and BC_5_Nax_2_ plants, whereas at the other salt concentrations it was significantly lower than in the control (Figure 6B). In cv. ‘Tamaroi’, increased POX activity was observed under all NaCl treatments.

#### 2.2.10. Hydrogen Peroxide (H_2_O_2_) Content

The hydrogen peroxide content of SMH87 plants was significantly higher than in the control ones only under 150 mM NaCl treatment. In cv. ‘Tamaroi’, the growing doses of NaCl increased the production of H_2_O_2_, and at 125 mM NaCl its content was 7.32-fold higher than in the control plants. The same relationship was noted in the BC_5_Nax_2_ line but then the H_2_O_2_ content was 2.5-fold higher than in the control.

#### 2.2.11. Yield Parameters

The salt treatments reduced the number of seeds per spike in all studied accessions (Figure 7A). The most drastic reduction was visible in the cv. ‘Tamaroi’, where a decrease of 23%, 40%, and 59% was observed at 100, 125, and 150 mM NaCl, respectively. The lowest reduction in the number of seeds per spike was shown by plants of the BC_5_Nax_2_ line at 150 mM NaCl; 32.3% fewer grains per spike were observed as compared with the control seed number. Salinity also had a negative impact on the dry weight (DW) of grain calculated per spike (Figure 7B). A decrease in seed DW along with the increasing NaCl concentration was observed. Drastic reduction in seed DW per spike was determined in cv. ‘Tamaroi’ at 100 mM NaCl, and amounted to 59.3% of the control. The higher NaCl concentrations declined grain DW by 67–75% as compared with the control. The mass of one thousand seeds (MTS) also decreased under increasing salinity. In the SMH87 line, the greatest decline in MTS was noted at 125 and 150 mM NaCl, and it was 35% and 45%, respectively, of the control. In cv. ‘Tamaroi’, a drastic 41% drop in MTS was already noted at 100 mM NaCl. At 125 and 150 mM NaCl, MTS significantly decreased by respectively 57% and 67% in relation to the control. BC_5_Nax_2_ plants showed the smallest loss in MTS in salt-treated soil. Moreover, at 125 and 150 mM NaCl, no significant differences between MTS were observed in this line. 

### 2.3. Correlation Analysis

A negative correlation between Ci and A, and a positive correlation between E and g_s_ and other gas exchange parameters were found (Table 2).

H_2_O_2_ content positively correlated with CAT and POX only in cv. ‘Tamaroi’ and BC_5_Nax_2_ line plants (Table 3). 

A negative correlation between TPC and CAT activity in all examined accessions was detected (Table 4). In all accessions, TPC positively correlated with SOD activity. The content of H_2_O_2_ correlated negatively with TPC only in cv. ‘Tamaroi’. No correlation between H_2_O_2_ and POX in all accessions was found.

## 3. Discussion

### 3.1. Experiment 1

#### 3.1.1. Percentage of Germinating Seeds, Germination Vigor, and Cell Membrane Permeability

Seedlings are the plant developmental stage the most sensitive to stresses occurring at the air–soil level. These stresses include, i.e., salinity and drought. An important aspect of salinity tolerance is the possibility of seeds germinating in saline soil and their ability to continue development. In their research, Płażek et al. [43] investigated the salinity tolerance of four wheat accessions. They found a difference in seed germination ability in common wheat cultivars in the saline soil. Contrary to that, two durum wheat accessions did not differ in the percentage of germinating seeds and they germinated even at 250 mM NaCl. However, the coleoptile length of both wheat species definitely decreased already at 70 mM NaCl. The authors concluded that salinity tolerance did not depend on wheat species but on their genotype. 

In the present experiment, we also observed significant differences in seed germination ability under salinity for all studied accessions. The lowest percentage of germinated seeds (PGS) was recorded in the most sensitive to salinity cv. ‘Tamaroi’, whereas the highest germination capacity was observed in the salt-resistant BC_5_Nax_2_ line. In the SMH87 line, the PGS decreased with increasing soil salinity. A similar relationship was demonstrated for CL and *Vi*. Brini et al. [3] also reported reduced PGS of durum wheat at 200 mM NaCl. Borlu et al. [44] found a large diversity in durum wheat cultivars germinated during exposure to 0 to 200 mM NaCl. Datir et al. [45] published similar findings for common wheat.

The electrolyte leakage is a common method used to evaluate cell membrane permeability under various stresses, i.e., under salinity [46]. In this study, the EL from the seeds of all accessions increased significantly with increasing salinity. Similar results were obtained by Płażek et al. [43] for two *Triticum durum* accessions. This parameter most strongly differentiates the response of durum wheat genotypes to salinity treatment. A positive correlation between CL, *Vi*, and PGS, and a negative correlation between CL, SSI, and EL were found. The study of Płażek et al. [43] yielded similar results. In our research, the parameters that most strongly differentiated the degree of salt sensitivity in durum wheat accessions from 100 mM NaCl upwards were the percentage of germinated seeds and germination vigor. Borlu et al. [44] reported considerable differences already at 75 mM NaCl. One of the reasons for the reduction in coleoptile growth under salinity could be combined osmotic-oxidative-toxic stress, which causes a disturbance of cell division, and modification of the structure of cell organelles [47]. In addition, salt causes the effect of plasmolysis, which is manifested by reduced turgor pressure, which compresses the cell cytoplasm resulting in a disturbance of the growth and shape of the emerging organs. Optimum cell hydration is essential for cell growth and its division [48].

Durum wheat is generally considered more sensitive to salinity than common wheat [49]. However, the results obtained by Płażek et al. [43] indicated that durum wheat was more tolerant to salinity at the germination stage, whereas common wheat showed better salt tolerance at the seedling stage. The salt susceptibility index (SSI) is a general parameter used for plant phenotyping as well as for determining the other components of physiological traits that provide information on the plant sensitivity or tolerance to stresses [41,42]. From an agricultural perspective, SSI is an important parameter taking into account the agronomic performance of the genotype under stress in relation to its yield under non-stress conditions [43]. In the present research, the highest values of SSI were recorded for salt-sensitive cv. ‘Tamaroi’ at both 125 and 150 mM NaCl. Płażek et al. [36] also observed an increase in SSI in durum wheat under increasing NaCl concentration, whereas a significant increase in this parameter was already observed at 70 mM NaCl. Thus, it can be concluded that the increased SSI reflects greater sensitivity to salinity.

### 3.2. Experiment 2

#### 3.2.1. Photosynthetic Efficiency

Kalaji and Pietkiewicz [44] reported that the analysis of chlorophyll fluorescence provides a quick insight into the photochemical efficiency of plants grown under different field conditions. Most of the examined parameters of ChlF significantly decreased due to increasing soil salinity. The parameters that most strongly differentiated the studied accessions under salinity were TR_o_/CS_m_, ET_o_/CS_m_, and DI_o_/CS_m_. The SMH87 line showed the largest decrease in the excitation energy trapped in PSII (TR_o_/CS_m_) and energy used for electron transport (ET_o_/CS_m_) at 150 mM. The most interesting result was a significant increase in the energy dissipation of PSII (DI_o_/CS_m_) only in the case of the SMH87 line at 150 mM. This result suggests that higher DI_o_/CS_m_, in this case, was an effect of a significant decrease in ET_o_/CS_m_ and TR_o_/CS_m_. The reduction in ET_o_/CS_m_ may be related, among other things, to the inactivation of the enzymatic complex responsible for water photodissociation. In addition, the F_v_/F_0_, maximum efficiency of water-splitting reaction at the donor side of PSII in SMH87 plants was almost two-fold lower at 150 mM than in the control and was significantly lower than in the other studied accessions. Therefore, it could be assumed that a low value of this parameter, as well as very high DI_o_/CS_m_, indicated a relatively low plant tolerance to salinity at the heading phase. Moradi and Ismail [50] showed that tolerance to salinity can be determined by higher dissipation of excess energy. According to Kalaji et al. [51], an increase in the dissipation of excessive energy could also indicate damage to the photosynthetic apparatus. These two statements indicate a difficult interpretation of changes in this parameter under stress conditions and suggest that it should be analyzed together with other parameters concerning energy absorption and electron transport within PSII. In SMH87 and BC_5_Nax_2_ plants, the performance index (PI) was the highest at 125 mM NaCl, whereas at 150 mM in the Polish SMH87 line it was higher than in cv. ‘Tamaroi’ and BC_5_Nax_2_ plants. These results do not clearly confirm if PI can be used as a salinity tolerance marker. A similar outcome was obtained by Płażek et al. [43]. However, Kalaji et al. [51] used PI to predict the seed yield of barley seven days after salt stress application. Płażek et al. [52] reported that PI turned out to be an excellent indicator of higher yield for narrow-leaf lupine germinating at low temperatures. It is worth noting that this parameter was measured at the seedling phase, much earlier than the seeds were harvested.

The photosynthesis rate depends on stomatal conductance and the availability of carbon dioxide [20]. In C4 plants or under stress conditions, mainly under osmotic stress, photosynthesis proceeds despite the closing of the stomata, caused by the plant’s defense response to the loss of turgor pressure. Carbonic anhydrase plays a key role in the photosynthesis rate through its effects on CO_2_ diffusion and other processes in photosynthetic organisms. It is mainly known to catalyze the CO_2_ to HCO_3_^−^ equilibrium. This reversible conversion has a clear role in sustaining the CO_2_ concentration at the site of ribulose-1,5-bisphosphate carboxylase/oxygenase (RuBisCO) [53]. In C3 plants, stomatal closure is recognized as a major protective mechanism against osmotic stress that decreases CO_2_ availability and photosynthetic activity. The lack of clear reduction in photosynthetic efficiency observed in durum wheat plants with closed stomata may suggest a significant use of C4-type primarily carboxylation via phosphoenolpyruvate carboxylase (PEPC) [54].

In terms of gas exchange intensity, the studied accessions responded specifically to rising salinity. The SMH87 line showed greater CO_2_ assimilation at all NaCl treatments than the control plants, whereas in cv. ‘Tamaroi’ and the BC_5_Nax_2_ line, the lowest *A* value was determined at 100 mM NaCl, and at higher salinity, the photosynthetic efficiency increased. In cv. ‘Tamaroi’ plants the highest net photosynthetic rate was achieved by the plants at 150 mM NaCl. Although the net photosynthetic rate correlated positively with stomatal conductance (r = 0.656; *p* < 0.05), this correlation was not observed at 150 mM NaCl. It is very interesting that a higher assimilation rate was provided when stomata were more closed. Zeeshan et al. [55] observed *A* and *E* reduction in both salt-sensitive and tolerant common wheat. This fact can be interpreted as a plant response to the osmotic stress in order to limit transpiration and reduce the amount of intercellular CO_2_ (*Ci*) in the stomata. According to Kalaji et al. [46], the first stage of the salinity effect on the photosynthesis of barley plants is the closing of the stomata (a decrease in *g_s_* parameter) rather than a reduction in PSII activity. In our experiment, a strong correlation was found between the transpiration rate and stomatal conductance (r = 0.909; *p* < 0.05). A decrease in transpiration rate may be a positive mechanism that can help to conserve water and reduce salt loading with the transpiration flux in the plant [56]. In the SMH87 line, the transpiration rate did not decrease at 150 mM NaCl, similarly to the net photosynthetic rate. In C3 plants, stomatal closure is recognized as a major protective mechanism against osmotic stress that decreases CO_2_ availability and photosynthetic activity. Non-reduction in photosynthetic efficiency with closed stomata in durum wheat plants may suggest a similarity to C4 plants, which can cope much better with stomatal closure, thanks to phosphoenolpyruvate carboxylase (PEPC) [50]. A weak but significant negative correlation was detected between CO_2_ concentration in the stomata (*Ci*) and the net photosynthesis rate (*A*) (r = −0.195; *p* < 0.05). This result suggests that in the studied accessions of durum wheat, CO_2_ from photorespiration could be used in the photosynthesis process. According to Garcia et al. [57] (2019), alternative carbon sources are important under stressful situations that reduce the uptake of atmospheric CO_2_ due to partial stomatal closure respiration and photorespiration stimulates internal sources of CO_2_.

Zeeshan et al. [55] compared two common wheat genotypes (salt-tolerant and salt-sensitive). Chlorophyll content decreased in the salt-sensitive genotype due to increasing salinity, whereas the net photosynthesis rate and intercellular CO_2_ concentration decreased significantly in both genotypes. Stomatal conductivity was also drastically reduced. Saqib et al. [58] observed a decrease in all gas exchange parameters at 150 mM NaCl in both resistant and sensitive common wheat genotypes. In our study, the gas exchange parameters did not differentiate the studied genotypes.

According to Bose et al. [59], a high concentration of carotenoids may protect plants from ROS damage during various environmental stresses. The SMH87 line showed an increase in Chl *a* content with increasing salinity, whereas the other accessions showed the opposite tendency; however, BC_5_Nax_2_ plants had the highest Chl *a* and *b* content at 150 mM NaCl. A similar effect in this accession was observed for the carotenoid content. It is interesting that in these salt-resistant plants the content of the photosynthetic pigments was the highest at the highest NaCl concentration. Zheng et al. [60] showed that under increasing salinity, the content of chlorophyll and carotenoids decreased more in salt-sensitive genotypes of common wheat. In our study, the content of photosynthetic pigments was also a good parameter differentiating the degree of salinity sensitivity of the studied accessions.

Total carbohydrate content in the flag leaf of all studied accessions, regardless of salt sensitivity degree, was higher than in the control. A different result was published by Azizpour et al. [61] in an experiment on durum wheat genotypes treated with salt in the range of 0–200 mM NaCl. Geissler et al. [55] did not find an increase in carbohydrate content in the leaves of a halophyte Aster pripolium in any salt treatment. Based on these results, it can be supposed that carbohydrate content in the leaves is not a valid parameter differentiating plants in terms of their resistance to salinity.

#### 3.2.2. Antioxidant System Phenolic Content, Antioxidant Enzymes, and H_2_O_2_

Abiotic stresses as high light, drought, salinity, anoxia, heavy metal ions induce plant immune responses through increased ROS production. The antioxidant compounds include enzymes as well as low-molecular phenolic compounds, e.g., flavonoids. [20,55,59,62,63,64]. The role of these compounds is ROS detoxification and protecting the organic compounds such as nucleic acids, proteins, and lipids. The synthesis of phenolics is generally induced in plants as a response to various stresses including salinity [56,57]. A reduced phenolic content was observed in Cynara cardunculus leaves under saline conditions [58]. According to Bose at al. [59], polyphenols are the key antioxidants limiting ROS damage in halophytes. Sharma et al. [65] stated that polyphenol oxidase activity is the strongest in salt-sensitive wheat and barley cultivars, intermediate in salt-tolerant genotypes, and the weakest in halophytes. These authors suggest that salt stress induces tissue damage in glycophytes but not in halophytes. In our study, the content of phenolics was also significantly affected by salt stress.

Low molecular weight antioxidants include mainly phenolic compounds [66]. In our work, we studied the content of total phenolic compounds (TPC) and cell wall-bound phenolic compounds (CWP). The latter plays a role in the plant’s response to various stresses by strengthening the cell wall or preventing excessive water leakage. Unfortunately, the content of both types of phenolic compounds did not significantly differentiate the studied durum wheat accessions in terms of salinity tolerance. In the leaves of all accessions, TPC was reduced due to salinity, whereas in terms of CWP each accession responded specifically. Ashraf et al. [67] observed that in the salt-sensitive genotype of common wheat, phenolic content decreased at 150 mM NaCl, as compared with that of the control.

Overproduction of ROS in the organisms due to salinity stress contributes to the oxidation of proteins that play a signaling role in many metabolic processes. Studies on various crops, such as peas, rice, wheat, barley, and tomatoes confirm that high soil salinity contributes to oxidative stress [8,62,63]. ROS cause lipid peroxidation in cellular membranes, DNA damage, protein denaturation, carbohydrate oxidation, decrease pigment content, and inhibit enzymatic activity [64,65]. Sairam et al. [68] found that wheat accessions differing in their salinity tolerance show different activity of antioxidant enzymes. Drought and salinity stress may induce light-dependent inactivation of the primary photochemistry associated with PSII. One of the most important antioxidant enzymes is CAT, which scavenges H_2_O_2_ in peroxisomes. In our study, all accessions showed an increase in CAT activity under increased soil salinity. POX activity varied specifically for each accession, although in all studied genotypes the highest POX activity was observed at 125 mM. Other authors obtained similar results. A reduction in POX activity due to salinity was observed in *Raphanus sativus* by Muthukumarasamy et al. [69], and in *Catharanthus roseus* (L.) by Jaleel et al. [70]. Decreasing POX activity under increasing salinity may indicate that this enzyme does not play a key role in defense mechanisms against salinity. It should be noted that POX is a more sensitive enzyme, activated at much lower concentrations of H_2_O_2_ than CAT. Peroxidases are activated at micromolar and CAT at millimolar concentrations of H_2_O_2_ [71]. Datir et al. [45] also observed an increase in CAT activity in wheat plants, especially in the salt-tolerant cultivar. Dioniso-Sese and Tobit [72] and Jaleel et al. [70] reported that higher NaCl concentrations reduced the activity of SOD in rice and *Catharanthus roseus*. In our study, a decrease in SOD activity under NaCl was also observed. Latef [73] claimed that in the salt-sensitive wheat genotype, the activity of SOD decreased under increasing salinity but it intensified in the intermediate and more salt-tolerant genotypes. 

The concentration of H_2_O_2_ strongly differentiated the studied accessions. This result is opposite to that obtained by Sairam et al. [68], who reported an increase in H_2_O_2_ at 100 and 200 mM NaCl in wheat cultivars, regardless of their sensitivity to salinity. In the most salt-sensitive, cv. ‘Tamaroi’, at 125 mM NaCl, H_2_O_2_ concentration increased more than 7-fold in relation to the control, and was 2-fold higher than that recorded at the same salt concentration in the BC_5_Nax_2_ line, and 7-fold higher than in the SMH87 line. The rapid increase in the H_2_O_2_ level in cv. ‘Tamaroi’ leaves at 125 mM NaCl did not affect CAT activity, although for all NaCl doses a positive correlation between these parameters was confirmed. It could be speculated that the source of additional H_2_O_2_ in the flag leaf was photorespiration. Increased photorespiration may be a defensive response to the toxic impact of Na^2+^ and Cl^−^ on the photosystems. Voss et al. [74] suggested that photorespiration plays a major role in the regulation of redox homeostasis under drought and salinity. Hydrogen peroxide is recognized as a signal molecule taking part in, e.g., lignin synthesis and activation of transcription factors, so its high level is in fact required in defense response to stress [75]. However, its too high concentration can be dangerous for the cell. A very high concentration of H_2_O_2_ in cv. ‘Tamaroi’ and CAT activity at 125 mM NaCl probably indicated that the enzyme activity was too low to alleviate oxidative stress. Only in this cultivar, a negative correlation between H_2_O_2_ and TPC (r = −0.410; *p* < 0.05) was found, which may suggest that phenolic compounds were engaged in H_2_O_2_ scavenging. Despite the presence of two types of antioxidants, such high levels of hydrogen peroxide could cause oxidative damage. In conclusion, an extremely high concentration of H_2_O_2_ under salt stress, much higher than in the other accessions, may be an indicator of higher susceptibility to this stress.

### 3.3. Yield Parameters

In the studied accessions, all yield parameters decreased under increasing salinity. The greatest reduction in the yield parameters was observed in cv. ‘Tamaroi’, where a drastic drop was noted at 125 and 150 mM NaCl. The lowest loss in the grain yield was observed in the most salt-resistant BC_5_Nax_2_ line at all salt doses. Among all studied yield parameters, MTS was the most differentiating for the studied accessions in terms of salt tolerance. Yield limitation is a common effect of salt stress and it is observed in many crop plants [76]. James et al. [39] reported a smaller decrease in the grain yield of durum wheat in plants with *Nax_2_* locus than in the other genotypes, along with increasing NaCl concentration in the soil. Similar results for durum wheat were obtained by Husain et al. [77], whereas Poustini and Siosemardeh [78] described the same relationship in common wheat. Francois et al. [79] also found a significant grain yield reduction in durum wheat with an increasing salt concentration in the soil. Moreover, the decrease was much greater than in common wheat genotypes. 

## 4. Materials and Methods

### 4.1. Plant Material

The study was performed on three spring durum wheat (*Triticum turgidum* L. subsp. *durum* (Desf.) Husn.) accessions differing in their salt tolerance. They were two Australian accessions: the salt-sensitive cv. ‘Tamaroi’ and salt-resistant BC_5_Nax_2_ line obtained from Dr. Richard A. James from CSIRO Plant Industry, and a Polish SMH87 line obtained from Dr. Jarosław Bojarczuk from the Plant Breeding Centre in Smolice, Plant Breeding and Acclimatization Institute Group. 

### 4.2. Experimental Design

Plant response to salinity was examined in two separate experiments (Figure 8). The first experiment was carried out in laboratory conditions in Petri dishes, where the plant response was determined in the germination phase. The analyses included the percentage of germinated seeds (PGS), germination vigor index (*Vi*), electrolyte leakage from seeds (EL), coleoptile length (CL), and salt susceptibility index (*SSI*). The second experiment was performed under controlled greenhouse conditions on plants in the phase of a fully developed flag leaf (BBCH–39) [80,81,82]. The flag leaves were used for the following analyses: chlorophyll *a* fluorescence, chlorophyll *a*, *b*, and carotenoid content (Chl *a*, *b* and Car), gas exchange, total soluble carbohydrate content (TSC), total phenolic content (TPC), cell wall-bound phenolic content (CWP), the activity of antioxidant enzymes, such as catalase (CAT), peroxidase (POX), superoxide dismutase (SOD), and hydrogen peroxide content (H_2_O_2_). Moreover, yield parameters such as seed number per spike, seed dry weight per spike, and mass of one thousand seeds (MTS) were determined.

### 4.3. Experiment 1—Laboratory Conditions

#### 4.3.1. Percentage of Germinated Seeds (PGS), Germination Vigor Index (*Vi*), Coleoptile Length (CL), and Salt Susceptibility Index (SSI)

Seeds of each accession were surface sterilized with 70% ethanol for 1 min, rinsed three times for 2 min with sterile water, and placed into Petri dishes (∅ = 9 cm) with filter paper wetted with four sodium chloride (NaCl) solutions: 0 (control), 50, 100, 125, and 150 mM; (10 plates × 15 seeds × 4 NaCl solutions for each accession). The seeds were germinated in a growth chamber (ST 5 C Smart, Pol-Aura Aparatura, Wodzisław Śląski, Poland), at 25 °C without access to light. The percentage of germinated seeds (PGS) and germination vigor index (*Vi*) were evaluated two days after sowing in five replicates for each durum wheat accession and NaCl solution. *Vi* was evaluated on the basis of a visual scale of coleoptile length (CL) according to Płażek et al. [43].

A visual scale was used, where 0—no germination; 1—coleoptile length of 1 mm; 2—coleoptile length of 2–3 mm; 3—coleoptile length of 4–7 mm; and 4—coleoptile length greater than 7 mm. The *Vi* index was calculated according to the formula:$$Vi=\left(n_{0}\times 0+\cdots +n_{4}\times 4\right)/N$$
where: n_x_—number of seeds assigned to a given coleoptile length; N—total number of seeds in the dish. *Vi* was assessed in five replicates for each accession and NaCl treatment.

Based on coleoptile length, SSI was estimated six days after sowing, according to the method described by Płażek et al. [43]. Coleoptile length (CL) was measured in 20 replicates for each NaCl treatment and accession. The influence of NaCl treatment on the percentage of germinated seeds and coleoptile length was presented as SSI using the following formula:$$\mathrm{SSI}=(1-G_{1}/G_{2})\times 100$$
where: G_1_—seeds germinated in NaCl solution, G_2_—seeds germinated in water (control). SSI for coleoptile length was calculated in a similar way. The results shown are the means of two independent experiments performed at the same time. 

#### 4.3.2. Electrolyte Leakage (EL)

To determine the plasma membrane permeability of the germinated seeds, three two-day-old germinated seeds with visible 2–3 mm coleoptile were collected from each accession and NaCl treatment. Next, the seeds were washed in distilled water and put into plastic vials containing 10 cm^3^ of ultra-pure water and shaken for 24 h at 150 rpm at room temperature. After that, electrical conductivity (*EL*_1_) was measured using a conductometer (CI 317, Elmetron, Zabrze, Poland), and then the samples were frozen at −80 °C for 24 h to achieve complete tissue degradation and to release all electrolytes. After 24 h, the samples were thawed and shaken again prior to the second measurement (*EL*_2_). Electrolyte leakage from the seeds was expressed as a percentage of total EL according to the formula: $$EL=\left(EL_{1}\times 100\right)/EL_{2}\;$$

All the measurements were performed in 10 replicates for each accession and NaCl treatment.

### 4.4. Experiment 2—Greenhouse Conditions

#### 4.4.1. Plant Cultivation

The seeds from each accession were sown into plastic pots (20 × 20 × 25 cm; five seeds per pot), in ten replicates (pots) for each accession and NaCl treatment (0, 100, 125, 150 mM). The plants were cultivated in commercial soil substrate, pH 5.8 (Ekoziem, Jurkow, Poland), mixed with sand (1:1; *v*/*v*). They were watered every day with the same volume of NaCl solution (300 cm^3^), and once a week with Hoagland’s medium [83]. Before sowing, the seeds were sterilized according to the procedure described in Experiment 1. The plants were grown until seed maturity in a greenhouse (50° 04′ 10.195′′ N, 19° 50′ 44.763′′ E) at 22/20 ± 1 °C (day/night), in daylight (May–August). To determine the actual NaCl concentration in the soil, the soil conductivity was estimated for each NaCl treatment according to the method described by Płażek et al. [43]. The results of the soil conductivity test for each NaCl treatment are presented in Table 5.

#### 4.4.2. Measurements of Chlorophyll *a* Fluorescence (ChlF)

Chlorophyll a fluorescence measurements were done with a plant efficiency analyzer (PEA; Hansatech Ltd., King’s Lynn, UK). The measurements involved the flag leaves after 25 min of adaptation to darkness (clips with a hole 4 mm in diameter). Before the measurements, the LED light source of a fluorimeter was calibrated using an SQS light meter (Hansatech Ltd., King’s Lynn, UK). Excitation irradiance had an intensity of 3 μmol m^−2^ s^−1^ (peak at 650 nm). Changes in fluorescence were recorded during irradiation between 10 μs and 1 s. During the initial 2 ms, the data were collected every 10 μs with 12-bit resolution. After this period, the frequency of measurements was reduced automatically. The data were used to calculate the following parameters based on the theory of energy flow in PSII and the JIP test [84,85]: ABS/CS_m_—energy absorption by antennas, TR_o_/CS_m_—excitation energy trapped in PS II, ET_o_/CS_m—_energy used for electron transport, DI_o_/CS_m—_energy dissipation from PS II, RC/CS_m_—number of active reaction centers, PI—performance index of PS II, F_v_/F_0_—maximum efficiency of water-splitting reaction of the donor side of PSII. All measurements were taken in ten replicates for each accession and NaCl treatment. 

#### 4.4.3. Measurements of Gas Exchanges Parameters

Gas exchange parameters measured in the flag leaves included photosynthesis rate (*A*), transpiration rate (*E*), stomatal conductance (*g_s_*), and intercellular CO_2_ concentration (*C_i_*). The analyses were performed with a CIRAS-3 infrared gas analyzer (PP Systems, Amesbury, MA, USA), with a Parkinson leaf chamber (PLC6). The irradiation system consisting of halogen lamps was applied. The airflow rate with a constant CO_2_ concentration of 400 μmol mol^−1^ through the assimilation chamber was 300 cm^3^ min^−1^. The temperature of the leaves was 22 °C, the air humidity was 40%, and the irradiance was 800 μmol photon m^−2^ s^−1^. To ensure the same conditions, all measurements were carried out between 9 am and 12 pm. All the measurements were taken in eight replicates for each accession and NaCl treatment.

#### 4.4.4. Chlorophyll (*a*, *b*) and Carotenoid (Car) Content

The measurements of chlorophyll a, b, and carotenoid content were performed in the flag leaves using the method described by Czyczyło-Mysza et al. [86] with a 96-well plate spectrophotometer (Synergy II, Biotek, Winooski, VT, USA). Absorbance was read at 470, 648, and 664 nm, and the total concentration of the measured pigments was calculated according to the formulas described by Lichtenthaler and Buschman [87]: Chl*a* (μg/cm^3^) = 13.36 A_664_ − 5.19 A_648_
Chl*b* (μg/cm^3^) = 27.43 A_648_ − 8.12 A_664_
Car (μg/cm^3^) = (1000 A_470_ − 2.13 Chla − 97.64 Chlb)/209
where: Chl*a*—chlorophyll *a*, Chl*b*—chlorophyll *b*, A_470_—absorbance at 470 nm, A_664_—absorbance at 664 nm, A_648_—absorbance at 648 nm. All the measurements were taken in three biological replicates for each accession and NaCl treatment.

#### 4.4.5. Total Soluble Carbohydrate Content (TSC)

Total water-soluble carbohydrate content was analyzed in the flag leaves by the phenol-sulfuric method according to Dubois et al. [88], with the modification described by Bach et al. [89]. Carbohydrate estimation was done in the samples collected from the flag leaves and prepared as described in Section 4.4.3. To this end, 10 μL of the extract was diluted with water to 200 μL, 200 cm^3^ of 5% phenol solution (*w*/*w*) was added, and 1 cm^3^ of concentrated H_2_SO_4_ was dispensed. Then the samples were vortexed and incubated for 20 min at room temperature and transferred to 96-well plates. Absorbance was read spectrophotometrically at 490 nm (Synergy II, Biotek, Winooski, VT, USA). The sugar content was finally calculated using a calibration curve where a glucose solution was used as a calibration standard. All the measurements were taken in three replicates for each accession and NaCl treatment. 

#### 4.4.6. Total Phenolic Compound Content (TPC)

Estimation of total phenolic compounds was performed in the flag leaves using a method described by Singleton et al. [90], with a modification from Bach et al. [89]. To the extracts prepared as described in Section 4.4.3. A Folin–Ciocalteu phenol reagent diluted with water (5:2, *v*/*v*) was added and left for 10 min. Then, saturated sodium carbonate (c.a. 25% *w*/*w*) was added. The ratio of these compounds in the samples was 100/400/400 µL (*v*/*v*/*v*). Next, the samples were incubated for 2 h at room temperature in the dark. After that, they were centrifuged (21,000× *g*, for 15 min at 15 °C) and transferred to 96-well plates. The absorbance was read spectrophotometrically at 760 nm (Synergy II, Biotek, Winooski, VT, USA). All measurements were taken in three replicates for each accession and NaCl treatment.

#### 4.4.7. Cell Wall-Bound Phenolic Content (CWP)

Content of CWP was assessed in the flag leaves according to Hura et al. [91]. The pellets obtained after pigment extraction (Section 4.4.3) were rinsed with 99.8% ethanol. Then, the samples were hydrolyzed with 3 M NaOH at room temperature overnight. Subsequently, concentrated HCl was added for the sample neutralization, and the samples were diluted with 1 cm^3^ of ethanol. The resulting solutions were analyzed for released phenolics similarly to soluble forms, as described in Section 4.4.6. All the measurements were taken in three replicates for each accession and NaCl treatment.

#### 4.4.8. Activity of Antioxidant Enzymes

Fresh plant material was collected from the flag leaves and homogenized in a hand mortar with 50 mM phosphate-potassium buffer (pH = 7.0) containing 0.1 mM EDTA (100 mg of FW plant material per 1 cm^3^ of the buffer). After centrifugation (10,000× *g*, 15 min at 4 °C, 32R, Hettich, Germany), the supernatant was subsampled and transferred to a 96-well plate format (Synergy II, Biotek, Winooski, VT, USA). The activity of superoxide dismutase (SOD, EC 1.15.1.1) was determined by the cytochrome reduction method according to McCord and Fridovic [92]. Catalase activity (CAT, E.C. 1.11.1.6) was measured at 240 nm according to the Aebi [93] method with H_2_O_2_ as a substrate. Peroxidase activity (POX, EC 1.11.1.7) was determined using the Lück [94] method with *p*–phenylenediamine as a substrate and absorbance was read at 485 nm. The analyses were conducted as described by Grudys et al. and references are cited therein [95,96,97]. The enzyme activities were calculated into protein content. The protein was analyzed using the Bradford [98] method. All the measurements were taken in three replicates for each accession and NaCl treatment.

#### 4.4.9. Hydrogen Peroxide Content (H_2_O_2_)

The method of plant material homogenization was the same as for the enzyme activity analyses (Section 4.4.6). The homogenization was performed at 4 °C in 50 mM phosphate-potassium buffer (pH = 7.0) containing 0.1 mM EDTA. The content of hydrogen peroxide was assessed using a commercial Amplex Red (10-acetyl-3,7-dihydro-xyphenoxazine) reagent kit [99] from Invitrogen (Waltham, MA, USA), according to the method provided in the manufacturer’s manual [100]. Pre-prepared samples of plant material were diluted with the reaction buffer, and the working solution containing a fluorescence probe precursor (Amplex Red). Next, 0.2 U·cm^−3^ of horseradish peroxidase was added and the samples were incubated for 30 min. Then they were transferred to a 96-well plate format (Synergy II, Biotek, Winooski, VT, USA) and fluorescence was read at Ex/Em 530/590 nm. The results were quantitated based on a calibration curve made for H_2_O_2_. All the measurements were taken in three replicates for each accession and NaCl treatment.

#### 4.4.10. Yield Parameters

After the harvest, ripe seeds were collected and the yield parameters were evaluated. The number and dry weight (DW) of seeds per spike were calculated in 30 replicates, and then the mass of one thousand seeds (MTS) for each accession and NaCl treatment was evaluated.

### 4.5. Statistical Analyses

The experiments were arranged and performed with the application of a completely randomized design. The normal distribution of data was analyzed using the Shapiro–Wilk test. The two-way analysis of variance (ANOVA) and Duncan’s multiple range test (at *p* < 0.05) were performed using the statistical package Statistica 13.3 (Stat–Soft, Inc., Tulsa, OK, USA). The data were presented as means ± SE (standard error). Pearson’s correlation coefficients were assumed as statistically significant at *p* < 0.05. MS Excel 2016 was used for drawing figures.

## 5. Conclusions

1.The percentage of germinated seed (PGS) and the germination vigor (*Vi*) are the parameters most differentiating the durum wheat accessions in terms of salt tolerance in the germination phase.2.Chlorophyll fluorescence parameters, such as maximum efficiency of water-splitting reaction of the donor side of photosystem II (PSII)—F_v_/F_0_ and energy dissipation from PSII—DI_o_/CS_m_ can be used as non-invasive parameters differentiating durum wheat accessions in terms of salinity tolerance.3.Salinity has a negative impact on grain yield by reducing the number of seeds per spike and the mass of one thousand seeds (MTS). The latter can be used as the most suitable parameter for determining tolerance to salinity stress.4.The salt-resistant BC_5_Nax_2_ line is characterized by the highest percentage of germinated seeds (PGS) and germination vigor (*Vi*) at the germination stage, and the highest content of chlorophyll *a*, *b*, and carotenoids, and MTS at the heading stage.5.The salt-sensitive cv. ‘Tamaroi’ shows significantly higher hydrogen peroxide levels at 125 and 150 mM NaCl, which proves considerable oxidative damage caused by salinity stress.6.From among the examined parameters, we chose those that most effectively differentiate durum wheat genotypes in terms of their salinity tolerance. These results can be helpful for breeders in the selection of genotypes the most resistant to this stress.7.Future research will include the analysis of proline content, hormonal profile in leaves, and the content of elements, especially the Na^+^/K^+^ ratio in the durum wheat genotypes examined in this study under salt stress.

## Figures and Tables

**Figure 1 ijms-23-08397-f001:**
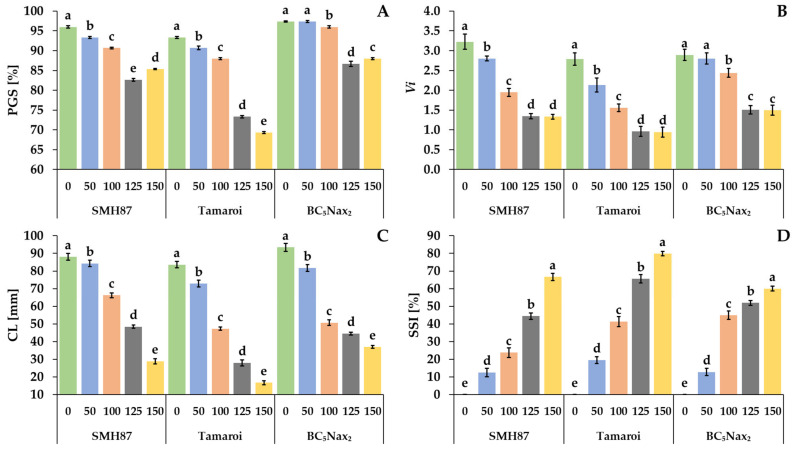
Effects of NaCl treatments (0, 50, 100, 125, 150 mM) on the PGS—percentage of germinated seeds (**A**), *Vi*—germination vigor index (**B**), CL—coleoptile length (**C**) and SSI—salt susceptibility index (**D**). The values represent means (n = 5) ± SE (standard error) for PGS and *Vi*; (n = 20) ± SE for CL and SSI within each accession. Mean values marked with the same letters do not differ statistically according to multiple range Duncan’s test (*p* < 0.05).

**Figure 2 ijms-23-08397-f002:**
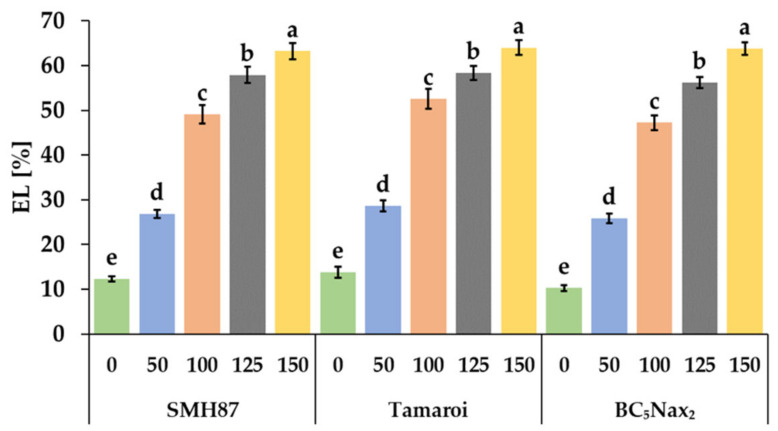
Effects of NaCl treatments (0, 50, 100, 125, 150 mM) on the electrolyte leakage (EL). The values represent means (n = 10) ± SE within each accession, and those marked with the same letters do not differ statistically according to multiple range Duncan’s test (*p* < 0.05).

**Figure 3 ijms-23-08397-f003:**
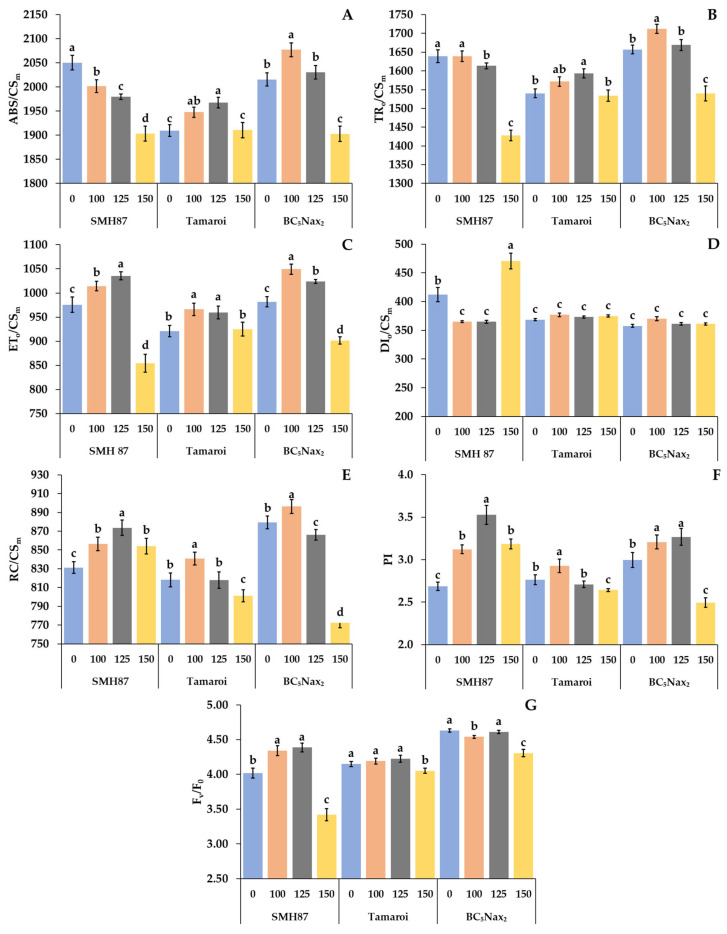
Effects of NaCl treatments (0, 100, 125, 150 mM) on chlorophyll *a* parameters: ABS/CS_m_—energy absorption by antennas (**A**); TR_o_/CS_m_—excitation energy trapped in PSII (**B**); ET_o_/CS_m_—energy used for electron transport (**C**); DI_o_/CS_m_—energy dissipation from PSII (**D**); RC/CS_m_—number of active reaction center (**E**); PI—performance index (**F**); F_v_/F_0_—maximum efficiency of water-splitting reaction of the donor side of PSII (**G**). The values represent means (n = 10) ± SE within each accession, and those marked with the same letters do not differ statistically according to multiple range Duncan’s test (*p* < 0.05).

**Figure 4 ijms-23-08397-f004:**
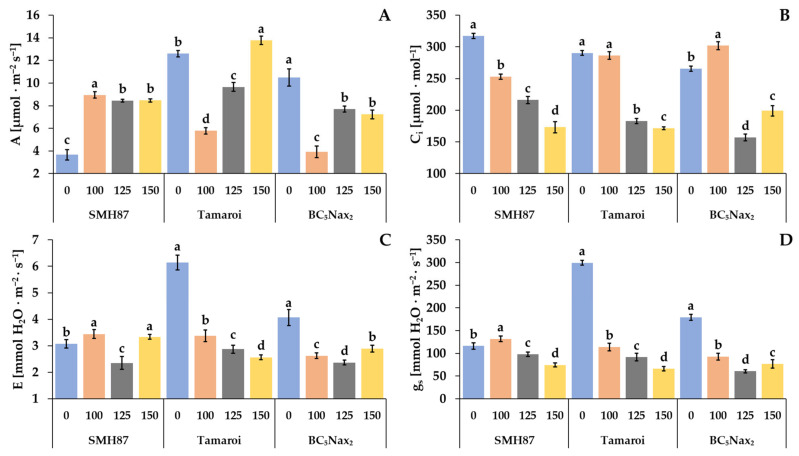
Effects of NaCl treatments (0, 100, 125, 150 mM) on the: *A*—net photosynthetic rate (**A**); *C_i_*—intercellular CO_2_ concentration (**B**); *E*—transpiration rate (**C**), *g_s_*—stomatal conductance (**D**). The values represent means (n = 8) ± SE within each accession, and those marked with the same letters do not differ statistically according to multiple range Duncan’s test (*p* < 0.05).

**Figure 5 ijms-23-08397-f005:**
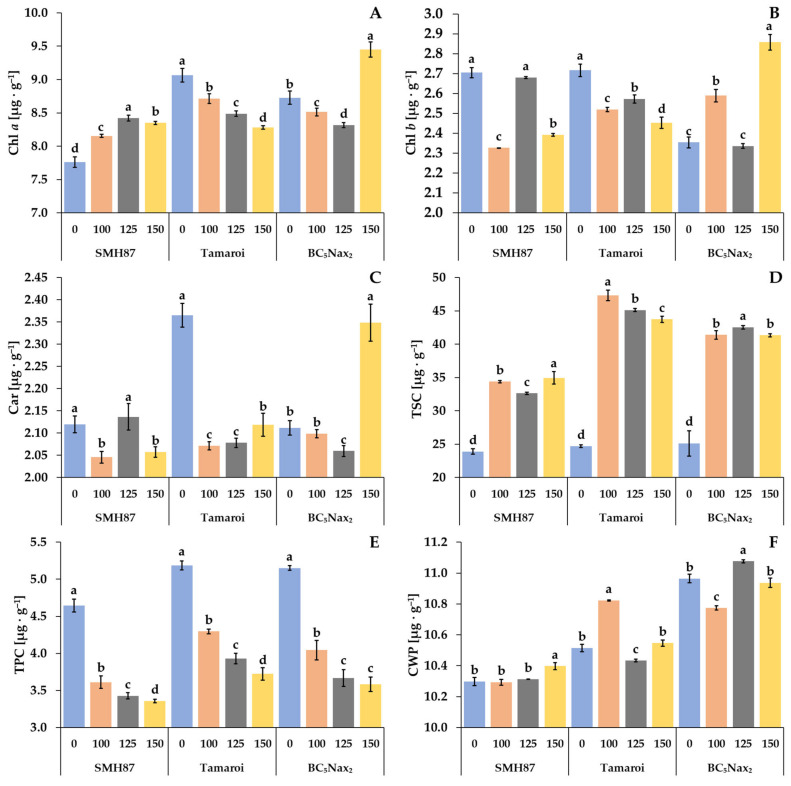
Effects of NaCl treatments (0, 100, 125, 150 mM) on Chl *a*—chlorophyll *a* (**A**), Chl *b*—chlorophyll *b* (**B**), Car—carotenoids (**C**), TSC—total soluble carbohydrates (**D**), TPC—total phenolic compounds (**E**), CWP—cell wall-bound phenolics (**F**). The values represent means (n = 3) ± SE within each accession, and those marked with the same letters do not differ statistically according to multiple range Duncan’s test (*p* < 0.05).

**Figure 6 ijms-23-08397-f006:**
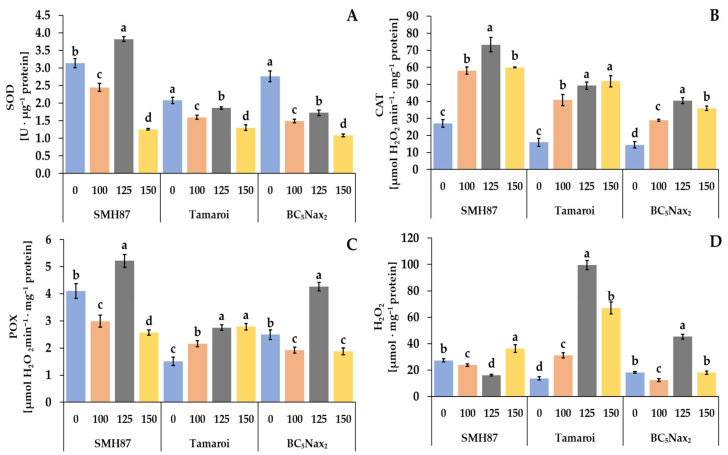
Effects of NaCl treatments (0, 100, 125, 150 mM) on SOD—superoxide dismutase (**A**), CAT—catalase (**B**), POX—peroxidase (**C**) activity, and H_2_O_2_—hydrogen peroxide (**D**) content. The values represent means (n = 3) ± SE within each accession, and those marked with the same letters do not differ statistically according to multiple range Duncan’s test (*p* < 0.05).

**Figure 7 ijms-23-08397-f007:**
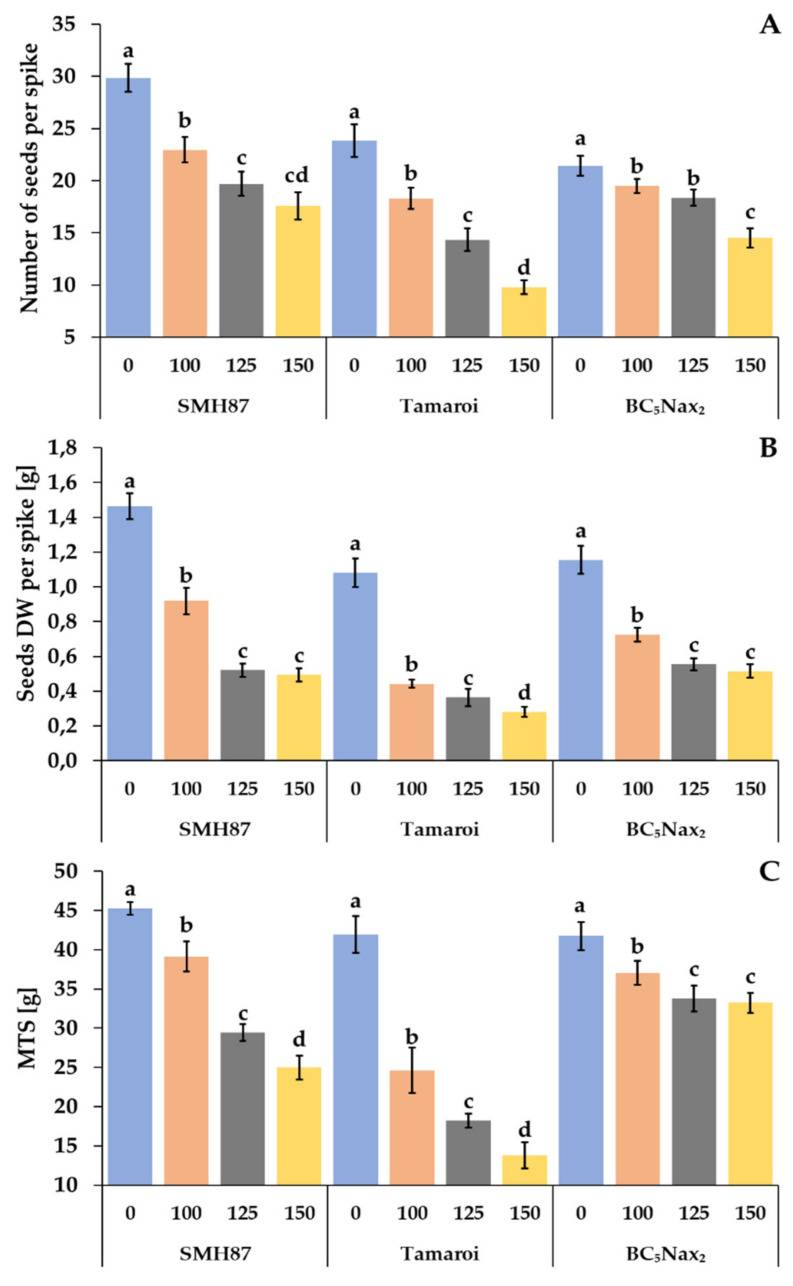
Effects of NaCl treatments (0, 100, 125, 150 mM) on yield components: number of seeds per spike (**A**), seed DW per spike (**B**), and MTS—mass of one thousand seeds (**C**). The values represent means (n = 3) ± SE within each accession, and those marked with the same letters do not differ statistically according to multiple range Duncan’s test (*p* < 0.05).

**Figure 8 ijms-23-08397-f008:**
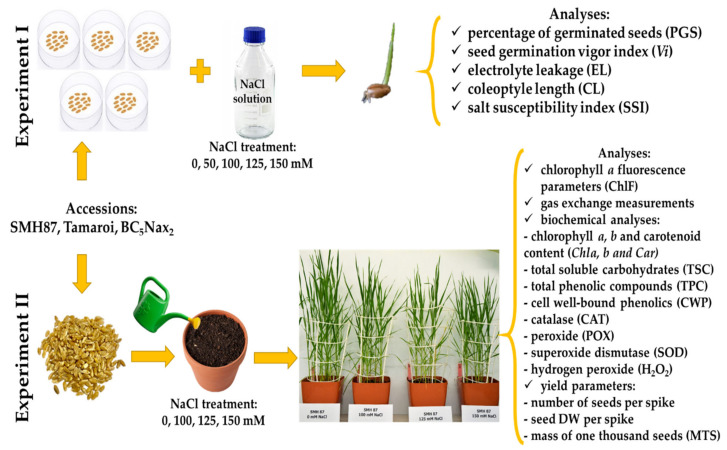
Design of the experiments.

**Table 1 ijms-23-08397-t001:** Pearson coefficients of linear correlation (*p* < 0.05) between CL—coleoptile length, SSI—salt susceptibility index, EL—electrolyte leakage, PGS—percentage of germinated seeds, *Vi*—vigor of germinating seeds.

Parameter	CL	SSI	EL	PGS
**SSI**	−0.954			
**EL**	−0.863	0.848		
**PGS**	0.63	−0.666	−0.572	
** *Vi* **	0.795	−0.822	−0.864	0.778

**Table 2 ijms-23-08397-t002:** Pearson coefficients of linear correlation (*p* < 0.05) for gas exchange parameters.

Parameter	*A*	*Ci*	*E*
** *Ci* **	−0.197		
** *E* **	0.664	0.408	
** *g_s_* **	0.656	0.509	0.909

*A*—net photosynthetic rate; *Ci*—intercellular CO_2_ concentration; *E*—transpiration rate, *g_s_*—stomatal conductance.

**Table 3 ijms-23-08397-t003:** Pearson coefficients of linear correlation (*p* < 0.05) between H_2_O_2_ and antioxidant enzyme activity in the leaves of three durum wheat accessions.

Accession	Parameter	CAT	POX	SOD
SMH87	H_2_O_2_	ns	ns	ns
Tamaroi		0.451	0.551	ns
BC_5_Nax_2_		0.640	0.712	ns

CAT—catalase; POX—peroxidase; SOD—superoxide dismutase; H_2_O_2_—hydrogen peroxide; ns—not significant.

**Table 4 ijms-23-08397-t004:** Pearson coefficients of linear correlation (*p* < 0.05) between TPC and antioxidant enzyme activity and H_2_O_2_ content in the leaves of three durum wheat accessions.

Accession	Parameter	CAT	POX	SOD	H_2_O_2_
SMH87	TPC	−0.884	ns	0.441	ns
Tamaroi		−0.533	ns	0.424	−0.410
BC_5_Nax_2_		−0.763	ns	0.553	ns

TPC—total phenolic compounds; CAT—catalase; POX—peroxidase; SOD—superoxide dismutase; H_2_O_2_—hydrogen peroxide; ns—not significant

**Table 5 ijms-23-08397-t005:** Conductivity of commercial soil watered with NaCl solution of 0, 100, 125, 150 mM.

NaCl (mM)	dS m^−1^·s^−1^
0	1.695
100	11.150
125	13.640
150	17.560

## Data Availability

All data relevant to the main findings of this study are included within the article.

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
