# Peer review of "Physiological and Biochemical Parameters of Salinity Resistance of Three Durum Wheat Genotypes"

_ijms, 2022, doi:10.3390/ijms23158397_

Round 1

Reviewer 1 Report

General comments

I have read the manuscript ijms-1814756). Entitle: Physiological and Biochemical Parameters of Salinity Resistance of Three Durum Wheat Genotypes written by Jakub Pastuszak et. al., for publication of ijms MDPI. In this study, the author investigates the physiological or biochemical parameters characterizing three durum wheat accessions with various tolerance to salinity. In this study author determine the electrolyte leakage (EL), and salt susceptibility index were determined the photosynthetic parameters, carbohydrate and phenolic content, antioxidant activity as well as wheat yield.

The overall research is very well conducted, and research is obvious application potential because this research author found that the salt-tolerant BC5Nax2 line was characterized by the highest PGS and Vi for NaCl concentration, highest content of chlorophyll a, b, and carotenoids. The most salt sensitive cv. 'Tamaroi' had many times higher H2O2 concentration. In this sense, the manuscript is much valuable. However, I found some points, especially the flow of the text and lack of potential references, and less concise the paragraphs, especially in the introduction and discussion sections. The author should provide enough examples and their interpretation of different traits of physiological and biochemical responses by the latest and appropriate references, some of which I mentioned below. Overall after I evaluate and request the author for this manuscript as a “MAJOR REVISION”.

  Major suggestions

1) introduction: In the second paragraph of the introduction author should mention the “salinity stress” as well as other abiotic stress background because abiotic stress (e.g., metal and drought) is the big figures of your study. For the metal stress and its related text please refer and cite this article “How plant copes with heavy metal” (doi:10.1186/1999-3110-55-35 ‎2014 and for the drought stress refer “Impact of drought stress on photosynthesis responses, leaf water potential and stem sap flow of two cultivars…. DOI:10.1016/j.scienta.2018.11.021 and mention the text “drought reduced the morphological traits such as reduction of leaf size and vegetative growth, and physiological traits such as reduction of photosynthesis and stomatal conductance and alter the stem anatomical features”

 2) Research hypothesis: Starting of the introduction of soil salinity stress, present scenario and its possible main factor which is much appreciated.  However, author should have focus the text salinity tolerance of wheat cultivars. Among these genotypes none of them not studies the salt tolerance previously? If yes author should also include those reference. If this is first work for these three Durum wheat then author should to be include the strong hypothesis of this research introduction because without appropriate hypotheses in introduction the entire introduction will be weaker. Please address this very clearly in the last section, such like why you are doing this and what is the future insight such text.

Some Line-by-Line comments

3) Line 13 (Abstract issue): Abstract should to be improved by minimize the descriptive sentences which is not suited in this section. Abstract should more logical, short, concise, and informative. Your abstract should reflect your study and major findings while shortly observed by readers. Please made the necessary corrections. Moreover, the last two-three lines (Ln- 26-30 is not very clear) please rephrase.

4) Line. 347 (Discussion section):

Discussion parts is well written, but I strongly suggest that the conciseness the text and whole manuscript. Good article always short and text should more concise. It may be good if author more descript about the better performance of physiological attributes and the seedling growth traits how it is linked to the Na and K discrimination in the discussion section in 3.1.1.

5) Line 394: Author should clearly present the discussion subsection 3.2.1 especially the stomatal and nonstomatal factor.  Author should include the information of stomatal and non-stomatal factor to determine the photosynthesis with enough reference paper. Generally, reduction of the pn due to cause of reduction of gs is the stomatal factor if internal CO2 concentration increased significantly that is caused due to non-stomatal factor.  https://doi.org/10.1016/j.envexpbot.2020.104111  This article well-describe these factors clearly. Refer and cite the article. I also strongly suggest to author to concise the text. Please delete the unnecessary description and less matching reference.

 6) Line no. 477 (Discussion): Author should properly cover the ROS and related text: ROS formation mainly for the detoxification under U stress or any other stress (light/metal/drought/flood). Refer to these two articles and cite those as a reference for section 2.5 in the discussion. (1) https://doi.org/10.1038/s41598-019-55889 (2) https://doi.org/10.1016/j.scitotenv.2021.146466 and mention somewhere in that paragraph “abiotic stress especially cause that the plant produces the ROS when the plant exposed to the stress condition and plant produce antioxidant, flavonoids, and secondary metabolites play to the role for protecting the plant for detoxifying ROS and protect the plant to protect the abnormal condition (i.e. stress) and protein and amino acid stabilization”

7) Line no. 739

Please write the detail information for the statistical text in the MM section. For e.g., which statistical software used for the figures? And which statistical package for e. g. version of SAS and its detail (e.g. SAS v. 9.4 (SAS Institute, Cary, NC, USA).

8) Line no. 746 (Conclusion)

Conclusion is comparatively weak. Author mostly repeat the same result part about the winter and spring resistance wheat and their characters very directly. Whole of your manuscript you already describe the resistance wheat obviously better performance the root and shoot characters as well as physiological perspective however conclusion should be some more solid not the result repetition. I agree some result might have repeated but the way of presentation should be change. Please alter the expression as possible. Actually, conclusions should be present the future insight of the research based on your current finding and strength of your results for the future research guideline of the related research. 

9) Line no. 784 (Reference): please double check the citations, its style and spell check and other grammatical errors. moreover, I request to authors for revision throughout the manuscript according to the journal rules.

 Good Luck!

Author Response

Thank you for your insightful review and valuable comments and for suggested valuable references. We tried to correct the text according to all your comments.

General comments

I have read the manuscript ijms-1814756). Entitle: Physiological and Biochemical Parameters of Salinity Resistance of Three Durum Wheat Genotypes written by Jakub Pastuszak et. al., for publication of IJMS MDPI. In this study, the author investigates the physiological or biochemical parameters characterizing three durum wheat accessions with various tolerance to salinity. In this study author determine the electrolyte leakage (EL), and salt susceptibility index were determined the photosynthetic parameters, carbohydrate and phenolic content, antioxidant activity as well as wheat yield. The overall research is very well conducted, and research is obvious application potential because this research author found that the salt-tolerant BC5Nax2 line was characterized by the highest PGS and Vi for NaCl concentration, highest content of chlorophyll a, b, and carotenoids. The most salt sensitive cv. 'Tamaroi' had many times higher H2O2 concentration. In this sense, the manuscript is much valuable. However, I found some points, especially the flow of the text and lack of potential references, and less concise the paragraphs, especially in the introduction and discussion sections. The author should provide enough examples and their interpretation of different traits of physiological and biochemical responses by the latest and appropriate references, some of which I mentioned below. Overall after I evaluate and request the author for this manuscript as a “MAJOR REVISION”.

Major suggestions

1) introduction: In the second paragraph of the introduction author should mention the “salinity stress” as well as other abiotic stress background because abiotic stress (e.g., metal and drought) is the big figures of your study. For the metal stress and its related text please refer and cite this article “How plant copes with heavy metal” (doi:10.1186/1999-3110-55-35 ‎2014 and for the drought stress refer “Impact of drought stress on photosynthesis responses, leaf water potential and stem sap flow of two cultivars…. DOI:10.1016/j.scienta.2018.11.021 and mention the text “drought reduced the morphological traits such as reduction of leaf size and vegetative growth, and physiological traits such as reduction of photosynthesis and stomatal conductance and alter the stem anatomical features”

Answer: It was done

 2) Research hypothesis: Starting of the introduction of soil salinity stress, present scenario and its possible main factor which is much appreciated. However, author should have focus the text salinity tolerance of wheat cultivars. Among these genotypes none of them not studies the salt tolerance previously? If yes author should also include those reference. If this is first work for these three Durum wheat then author should to be include the strong hypothesis of this research introduction because without appropriate hypotheses in introduction the entire introduction will be weaker. Please address this very clearly in the last section, such like why you are doing this and what is the future insight such text.

Answer: We followed the comments and supplemented the text with the suggested information and citations. The new text is marked in red.

Some Line-by-Line comments

3) Line 13 (Abstract issue): Abstract should to be improved by minimize the descriptive sentences which is not suited in this section. Abstract should more logical, short, concise, and informative. Your abstract should reflect your study and major findings while shortly observed by readers. Please made the necessary corrections. Moreover, the last two-three lines (Ln- 26-30 is not very clear) please rephrase.

Answer: The abstract was corrected and supplemented with more understandable conclusions.

 4) Line. 347 (Discussion section):

Discussion parts is well written, but I strongly suggest that the conciseness the text and whole manuscript. Good article always short and text should more concise. It may be good if author more descript about the better performance of physiological attributes and the seedling growth traits how it is linked to the Na and K discrimination in the discussion section in 3.1.1.

Answer: The text throughout the manuscript has been shortened where possible. In the discussion part 3.1.1. we supplemented the interpretations of the obtained results and referred to additional references. In the case of Na and K discrimination, we know that this parameter is very important in salinity tolerance, however, in this part of the research we described in the manuscript, we do not present the results of the analysis of elements in roots and shoots, but they will be presented in the next manuscript along with other parameters.

5) Line 394: Author should clearly present the discussion subsection 3.2.1 especially the stomatal and nonstomatal factor.  Author should include the information of stomatal and non-stomatal factor to determine the photosynthesis with enough reference paper. Generally, reduction of the pn due to cause of reduction of gs is the stomatal factor if internal CO2 concentration increased significantly that is caused due to non-stomatal factor.  https://doi.org/10.1016/j.envexpbot.2020.104111  This article well-describe these factors clearly. Refer and cite the article. I also strongly suggest to author to concise the text. Please delete the unnecessary description and less matching reference.

Answer: It was done

 6) Line no. 477 (Discussion): Author should properly cover the ROS and related text: ROS formation mainly for the detoxification under U stress or any other stress (light/metal/drought/flood). Refer to these two articles and cite those as a reference for section 2.5 in the discussion. (1) https://doi.org/10.1038/s41598-019-55889 (2) https://doi.org/10.1016/j.scitotenv.2021.146466 and mention somewhere in that paragraph “abiotic stress especially cause that the plant produces the ROS when the plant exposed to the stress condition and plant produce antioxidant, flavonoids, and secondary metabolites play to the role for protecting the plant for detoxifying ROS and protect the plant to protect the abnormal condition (i.e. stress) and protein and amino acid stabilization”

Answer: we added suggested references and corrected this part of text  

7) Line no. 739

Please write the detail information for the statistical text in the MM section. For e.g., which statistical software used for the figures? And which statistical package for e. g. version of SAS and its detail (e.g. SAS v. 9.4 (SAS Institute, Cary, NC, USA).

Answer: It was done 

8) Line no. 746 (Conclusion)

Conclusion is comparatively weak. Author mostly repeat the same result part about the winter and spring resistance wheat and their characters very directly. Whole of your manuscript you already describe the resistance wheat obviously better performance the root and shoot characters as well as physiological perspective however conclusion should be some more solid not the result repetition. I agree some result might have repeated but the way of presentation should be change. Please alter the expression as possible. Actually, conclusions should be present the future insight of the research based on your current finding and strength of your results for the future research guideline of the related research. 

Answer: Conclusions were corrected. We examined only spring durum wheat accessions and this information was added in material and methods.

9) Line no. 784 (Reference): please double check the citations, its style and spell check and other grammatical errors. moreover, I request to authors for revision throughout the manuscript according to the journal rules.

Answer: It was done 

Reviewer 2 Report

The presented Manuscript Physiological and Biochemical Parameters of Salinity Resistance of Three Durum Wheat Genotypes by Jakub Pastuszak, MichaÅ‚ Dziurka, Marta Hornyák, Anna Szczerba, PrzemysÅ‚aw Kopeć and Agnieszka PÅ‚ażek is a qualitatively performed study of the physiological biochemical response of plants to stress. The work was done with high quality and contains many characteristics related to respiration, the characteristics of photosynthesis. The disadvantages of the work are a certain one-sidedness in the discussion of the material and the lack of data on cytological and physiological data that characterize the development of the aerial parts of plants and the accumulation of starch and the provision of gravitropically given growth of the root system due to the columella of the cap. There are a number of modern works on this topic, and I think it is necessary to expand the discussion of metabolism: Terletskaya, N. V., Lee, T. E., Altayeva, N. A., Kudrina, N. O., Blavachinskaya, I. V., & Erezhetova, U. (2020). Some Mechanisms Modulating the Root Growth of Various Wheat Species under Osmotic-Stress Conditions. Plants, 9(11), 1545; Kononenko, N., Baranova, E., Dilovarova, T., Akanov, E., & Fedoreyeva, L. (2020). Oxidative damage to various root and shoot tissues of durum and soft wheat seedlings during salinity. Agriculture, 10(3), 55; Baranova, E. N., & Gulevich, A. A. (2021). Asymmetry of plant cell divisions under salt stress. Symmetry, 13(10), 1811; Ibrahimova, U., Suleymanova, Z., Brestic, M., Mammadov, A., Ali, O. M., Abdel Latef, A. A. H., & Hossain, A. (2021). Assessing the Adaptive Mechanisms of Two Bread Wheat (Triticum aestivum L.) Genotypes to Salinity Stress. Agronomy, 11(10), 1979. This especially concerns the features of growth disorders due to disorders of the cytoskeleton and structural organization of organelles associated with gene expression and changes in chromatin packaging due to oxidative stress causing damage under the action of Na salts: Baranova, E. N., & Gulevich, A. A. (2021). Asymmetry of plant cell divisions under salt stress. Symmetry, 13(10), 1811; Ding, H., Ma, D., Huang, X., Hou, J., Wang, C., Xie, Y., ... & Guo, T. (2019). Exogenous hydrogen sulfide alleviates salt stress by improving antioxidant defenses and the salt overly sensitive pathway in wheat seedlings. Acta Physiologiae Plantarum, 41(7), 1-11.

The reasons for not paying attention to the phenotype and condition of the stomata, cell size, and changes in the morphology and thickness of the leaf - parameters directly related to respiration and photosynthesis and the outflow of assimilates are not clear. They can be added only in a descriptive way, otherwise it is difficult to understand whether the isolated substances can be compared. In general, the manuscript is formatted according to the rules, except for the conclusion section, where numbering is used. What should be excluded or reformatted to 5.1...5.2...5.3 mode

Author Response

Thank you for review, for your time and for suggesting valuable references. Most of them were used in the respective chapters of the work.

The presented Manuscript Physiological and Biochemical Parameters of Salinity Resistance of Three Durum Wheat Genotypes by Jakub Pastuszak, MichaÅ‚ Dziurka, Marta Hornyák, Anna Szczerba, PrzemysÅ‚aw Kopeć and Agnieszka PÅ‚ażek is a qualitatively performed study of the physiological biochemical response of plants to stress. The work was done with high quality and contains many characteristics related to respiration, the characteristics of photosynthesis. The disadvantages of the work are a certain one-sidedness in the discussion of the material and the lack of data on cytological and physiological data that characterize the development of the aerial parts of plants and the accumulation of starch and the provision of gravitropically given growth of the root system due to the columella of the cap. There are a number of modern works on this topic, and I think it is necessary to expand the discussion of metabolism: Terletskaya, N. V., Lee, T. E., Altayeva, N. A., Kudrina, N. O., Blavachinskaya, I. V., & Erezhetova, U. (2020). Some Mechanisms Modulating the Root Growth of Various Wheat Species under Osmotic-Stress Conditions. Plants, 9(11), 1545; Kononenko, N., Baranova, E., Dilovarova, T., Akanov, E., & Fedoreyeva, L. (2020). Oxidative damage to various root and shoot tissues of durum and soft wheat seedlings during salinity. Agriculture, 10(3), 55; Baranova, E. N., & Gulevich, A. A. (2021). Asymmetry of plant cell divisions under salt stress. Symmetry, 13(10), 1811; Ibrahimova, U., Suleymanova, Z., Brestic, M., Mammadov, A., Ali, O. M., Abdel Latef, A. A. H., & Hossain, A. (2021). Assessing the Adaptive Mechanisms of Two Bread Wheat (Triticum aestivum L.) Genotypes to Salinity Stress. Agronomy, 11(10), 1979. This especially concerns the features of growth disorders due to disorders of the cytoskeleton and structural organization of organelles associated with gene expression and changes in chromatin packaging due to oxidative stress causing damage under the action of Na salts: Baranova, E. N., & Gulevich, A. A. (2021). Asymmetry of plant cell divisions under salt stress. Symmetry, 13(10), 1811; Ding, H., Ma, D., Huang, X., Hou, J., Wang, C., Xie, Y., ... & Guo, T. (2019). Exogenous hydrogen sulfide alleviates salt stress by improving antioxidant defenses and the salt overly sensitive pathway in wheat seedlings. Acta Physiologiae Plantarum, 41(7), 1-11.

The reasons for not paying attention to the phenotype and condition of the stomata, cell size, and changes in the morphology and thickness of the leaf - parameters directly related to respiration and photosynthesis and the outflow of assimilates are not clear. They can be added only in a descriptive way, otherwise it is difficult to understand whether the isolated substances can be compared. In general, the manuscript is formatted according to the rules, except for the conclusion section, where numbering is used. What should be excluded or reformatted to 5.1...5.2...5.3 mode

In our research, we did not plan the analysis of differences in the morphological and anatomical structure of the examined durum wheat genotypes, therefore we cannot refer to the above comments.

Regarding to the comparison of the content of compounds in different objects, we want to explain that: the content and activity of the analyzed compounds within one genotype at different salt concentrations were compared to that of the control plants, not treated with salt. Moreover, we compared the content of compounds or enzyme activity for three genotypes but within the same NaCl concentration.

In the part concerning conclusions, the same numerical breakdown as in other chapters of the manuscript is not used, for example: Results, Material and methods, hence in chapter 5. Conclusions, individual conclusions were numbered from 1-5.

Round 2

Reviewer 1 Report

Dear Author

I have read the revised manuscript (ijms-1814756). Titled: Physiological and Biochemical Parameters of Salinity Resistance of Three Durum Wheat Genotypes for publication in ijms MDPI. This is the second submission made by the author. The author addressed all the questions and suggestions that I raised the issue in the review of the original manuscript. I satisfy the author’s revisions throughout the paper. Author well addresses the abstract issues. Especially author improved the introduction and discussion section very well inflow. Now, this manuscript improved the flow of writing, which was comparatively shallow in the original version but in this revised copy author addressed all the quarries and suggestions very well. Before accepting this manuscript if there is anything needed to be revised by the author, especially English grammar, or spell check, I request this manuscript is currently in “Minor Revision” and any grammatical error author may improve in this stage. Thank you.

Reviewer 2 Report

Authors Physiological and Biochemical Parameters of Salinity Resistance of Three Durum Wheat Genotypes Manuscript by Jakub Pastuszak, MichaÅ‚ Dziurka, Marta Hornyak, Anna Szczerba, PrzemysÅ‚aw Kopec, Agnieszka PÅ‚azek contains a qualitative study examining salinity tolerance in a sodium chloride model. The text has been revised, edited and can be accepted for publication.